# Synthesis and Antimicrobial Activity of Novel 4-Hydroxy-2-quinolone Analogs

**DOI:** 10.3390/molecules25133059

**Published:** 2020-07-04

**Authors:** Thitiphong Khamkhenshorngphanuch, Kittipat Kulkraisri, Alongkorn Janjamratsaeng, Napasawan Plabutong, Arsa Thammahong, Kanitta Manadee, Sarisa Na Pombejra, Tanatorn Khotavivattana

**Affiliations:** 1Department of General Education, Faculty of Science and Health Technology, Navamindradhiraj University, Bangkok 10300, Thailand; thitiphong.kha@nmu.ac.th; 2Department of Chemistry, Faculty of Science, Chulalongkorn University, Bangkok 10330, Thailand; kittipatkulkraisri@gmail.com (K.K.); mekans89@gmail.com (A.J.); 3Antimicrobial Resistance and Stewardship Research Unit, Department of Microbiology, Faculty of Medicine, Chulalongkorn University, Bangkok 10330, Thailand; fah34405@gmail.com (N.P.); arsa.t@chula.ac.th (A.T.); 4Department of Microbiology, Faculty of Science, Chulalongkorn University, Bangkok 10330, Thailand; noonkanitta@gmail.com (K.M.); sarisa.n@chula.ac.th (S.N.P.); 5Center of Excellence in Natural Products Chemistry, Department of Chemistry, Faculty of Science, Chulalongkorn University, Bangkok 10330, Thailand

**Keywords:** antibacterial, antifungal, 4-hydroxy-2-quinolinone, structure–activity relationship

## Abstract

Alkyl quinolone has been proven to be a privileged scaffold in the antimicrobial drug discovery pipeline. In this study, a series of new 4-hydroxy-2-quinolinone analogs containing a long alkyl side chain at C-3 and a broad range of substituents on the C-6 and C-7 positions were synthesized. The antibacterial and antifungal activities of these analogs against *Staphylococcus aureus*, *Escherichia coli*, and *Aspergillus flavus* were investigated. The structure-activity relationship study revealed that the length of the alkyl chain, as well as the type of substituent, has a dramatic impact on the antimicrobial activities. Particularly, the brominated analogs **3j** with a nonyl side chain exhibited exceptional antifungal activities against *A. flavus* (half maximal inhibitory concentration (IC_50_) = 1.05 µg/mL), which surpassed that of the amphotericin B used as a positive control. The antibacterial activity against *S. aureus*, although not as potent, showed a similar trend to the antifungal activity. The data suggest that the 4-hydroxy-2-quinolone is a promising framework for the further development of new antimicrobial agents, especially for antifungal treatment.

## 1. Introduction

The infection produced by microorganisms, such as bacteria or fungi, has long been considered as one of the major global health concerns. Recently, the rapid emergence of drug-resistant pathogens has caused a dramatic decrease in the effectiveness of the currently available medications [1]. At present, antimicrobial resistance causes 700,000 deaths annually, and the number is expected to rise to 10 million in 2050 [2]. As a result, the development of a new class of antimicrobial drugs is highly in demand as a complement or replacement to existing antibiotics.

In the search for a new scaffold for antimicrobial drugs, 4-hydroxy-2-quinolone, a class of heterocyclic compound, has shown great potential, as various analogs of 4-hydroxy-2-quinolone including Roquinimex (Linomide), a synthetic immunomodulator [3], have been reported to possess a wide range of biological activities such as antibacterial [4], antifungal [5], anticancer [6], antioxidant [7], and so on. Over the past few decades, several attempts have been made to investigate the structure–activity relationship of the antimicrobial activity of this scaffold; for instance, the derivatives of 4-hydroxy-2-quinolone bearing chalcones [8], pyrimidines [9], pyrazoles [10], and thiadiazole [11] (Figure 1a). Despite the extensive studies on the substituent pattern at the C-3 position, especially with aromatic or heteroaromatic motifs, there is a limited number of studies on the effect of the aliphatic side chain on the antimicrobial activity. According to a study on a similar scaffold such as 4-hydroxycoumarin [12], or quinolone [13,14], it was found that the alkyl side chain plays a crucial role in the antibacterial activity as reflected by the strong correlation between the length of the chain and their bioactivity. In a previous study, it was found that the alkenyl side chain of 4-quinolones can form hydrophobic interactions at a binding pocket within MurE ligase of *Mycobacterium tuberculosis* [15]. However, the mechanism of action of 4-hydroxy-2-quinolone remains unclear. To date, the effect of substituents on the benzo-carbons of the 2-quinolone core structure has never been thoroughly investigated for the antimicrobial activity; only the 6-nitro [16], 6,7-dimethoxy [17], and 5-hydroxy and/or 6-carboxylic acid [18] analogs were investigated (Figure 1a). According to the development of ciprofloxacin, a 4-quinolone drug popularly used until today, it was found that the introduction of a fluorine atom at the C-6 position dramatically enhanced the antibacterial activity [19]. Moreover, further study revealed that the type of substituents on the C-7 position led to an enormous impact on their antimicrobial properties [20]. However, for the 4-hydroxy-2-quinolone core structure, the only thorough investigation of the structure–activity relationship on the benzo-carbons was reported by Jönsson and group in 2004, regarding their inhibitory effects on autoimmune disorders [21]. On the other hand, the structure–activity relationship for antimicrobial activity remains unclear. Herein, we report the diverted total synthesis of novel 4-hydroxy-2-quinolone analogs bearing an alkyl side chain at C-3 and various types of substituents on C-6 and C-7 positions (Figure 1b). The obtained compounds were then tested for their antibacterial and antifungal activities in order to obtain additional knowledge on the structure–activity relationship of this promising scaffold.

## 2. Results and Discussions

### 2.1. Chemical Synthesis

The 4-hydroxy-2-quinolone analogs in this work were synthesized following a three-step protocol starting from isatoic anhydrides with various substituents on the aromatic ring (Figure 2). First, the alkylation of isatoic anhydrides was performed using either methyl iodide or ethyl iodide as the alkylating agents and *N*,*N*-diisopropylethylamine (DIPEA) as a base, leading to the formation of *N*-alkylated isatoic anhydrides (**1a**–**1g**) in moderate to excellent yields [22]. Next, β-ketoesters (**2a** and **2b**) bearing long alkyl chains such as nonyl and tridecyl groups were synthesized by the reaction between ethyl potassium malonate and the appropriate acid chlorides mediated by MgCl_2_ and triethylamine in good yields [23]. Finally, the condensation between **1a**–**1g** and **2a** or **2b** was achieved using NaH as a base, yielding the analogs of 4-hydroxy-2-quinolone (**3a**–**3j**) in relatively low yields [24]. The structures of all the synthesized 4-hydroxy-2-quinolone analogs have been confirmed by several spectroscopic techniques such as ^1^H NMR, ^13^C NMR, IR, and HRMS. It is worth noting that these compounds show a characteristic pattern in ^13^C NMR: A signal at around δ 209 ppm indicating the carbonyl group on the side chain, and a set of signals at 174 and 161 ppm indicating C-4 and C-2 of the quinolone ring, respectively.

In order to gain insight into the reason behind the poor yield of the final step, the reaction between **1d** and **2a** for the synthesis of **3d** was thoroughly investigated (Figure 3). The isolation and characterization of byproducts revealed that, apart from the desired 4-hydroxy-2-quinolone analog **3d**, which were obtained in only 9%, two additional products also occurred: 1,4-Dihydroquinoline-3-carboxylate analog **4d**, which is a structural isomer of the desired product, and the 2-aminobenzoic acid **5d** were obtained in 17% and 51%, respectively. The generation of the two possible isomers, **3d** and **4d**, has been documented by Coppola and Hardtmann [25], demonstrating that the selectivity of such transformation highly depends on the nature of the methylene species, which is the β-ketoester **2a** in this case. The formation of a large amount of **5d** could be caused by the hydrolysis of the isatoic anhydride **1d**, hence indicating the presence of water in the reaction. Although several attempts were made to exclude the moisture from the reaction, the yield of **3d** was not significantly improved.

### 2.2. Antimicrobial Activity and Structure–Activity Relationship

All the synthesized 4-hydroxy-2-quinolone analogs **3a**–**3j**, as well as the isomer **4d**, were screened for their antifungal activity against a pathogenic fungus *Aspergillus flavus*, as well as the antibacterial activity against a Gram-positive bacteria *Staphylococcus aureus* and a Gram-negative bacteria *Escherichia coli* (Table 1). For the antifungal activity, compound **3a** with no substituents on the ring exhibited very low inhibition with a half maximal inhibitory concentration (IC_50_) of 70.97 ± 3.71 µg/mL. The introduction of the substituents on the benzo-carbons led to significant improvement in the potency, especially for the halogenated analogs **3d**, **3e**, and **3f** with up to fivefold enhanced activity. This result is in line with a previous study reported by Huigens and group that the introduction of halogen substituents, especially for bromine, to the quinoline core structure can lead to higher antimicrobial activity, probably due to the metal(II)-dependent mechanism [26]. 

Replacing the methyl group with the ethyl group at the position R^2^ (**3g**) led to a dramatic improvement in the inhibitory activity (IC_50_ = 9.48 ± 6.17 µg/mL). The same trend was observed for the replacement of the tridecyl with the nonyl group at the position R^3^ (**3h**) with the IC_50_ of 8.45 ± 1.59 µg/mL. This result suggests that an optimum size of these alkyl side chains is required, presumably to fit into the hydrophobic pockets within microbial enzymes, as demonstrated earlier by Bhakta and group [15]. Further enhancement in the antifungal activity was achieved when halogen substituents were introduced at the benzo-carbon positions on the nonyl analog. The IC_50_ values of the 6-chloro (**3i**) and 6-bromo (**3j**) nonyl analogs were 4.60 ± 1.14 and 1.05 ± 1.31 µg/mL, respectively, which is almost 70 times higher than that of **3a**. The inhibitory activity of **3j** against *A. flavus* is even greater than amphotericin B, the current fungicidal antifungal agent, with almost two times more effectiveness. Lastly, the 1,4-dihydroquinoline-3-carboxylate analog **4d** showed slightly higher activity compared to its corresponding isomer **3d**, which may suggest the possibility of further antifungal study on this scaffold. For the antibacterial activity, it seems like these 4-hydroxy-2-quinolone analogs possess almost no activity against both *S. aureus* and *E. coli*. Nevertheless, two compounds, **3i** and **3j**, exhibited significant inhibitory activity against *S. aureus*. This result correlates well with the antifungal activity in which these two analogs also showed the highest potency. Although the minimum inhibitory concentrations (MICs) of **3i** and **3j** were unclear, both compounds demonstrated inhibitory effects on the bacterial growth at concentrations between 125–1000 and 125–500 µg/mL, respectively. It was interesting to notice that the high concentration of compounds at 1000 µg/mL gave worse antibacterial results than the medium concentrations (125–500 µg/mL; see Appendix A). This phenomenon has been found in some quinolone-class antibiotics such as nalidixic acid. The survival of bacteria at high concentrations of quinolones was explained by a decrease in reactive oxygen species (ROS) level during exceeding quinolone concentrations [27]. The ROS accumulation is a key mechanism of quinolone-mediated lethality [28], and this may relate to the antibacterial activity of **3i** and **3j**. However, the result of minimum bactericidal concentration (MBC) indicated that both **3i** and **3j** exhibited a bacteriostatic effect rather than bactericidal activity (see Appendix A). Taken together, the data suggest that the synthesized 4-hydroxy-2-quinolone analogs could be further developed as promising antimicrobial agents especially for antifungal treatment. As cellular structures of fungi and bacteria are largely different, based on our antimicrobial results, these newly synthesized compounds may mainly target fungal molecules rather than bacterial ones. This seems to be their outstanding property that is not found in other quinolone drugs, which are primarily used as an antibacterial treatment.

## 3. Materials and Methods

### 3.1. General Experimental Procedures

All reagents and solvents were obtained from Sigma-Aldrich (St. Louis, MO, USA), TCI Chemicals (Tokyo, Japan), Fluorochem (Hadfield, Derbyshire, UK), and Merck (Darmstadt, Germany). All solvents for column chromatography from RCI Labscan (Samutsakorn, Thailand) were distilled before use. ^1^H and ^13^C NMR spectra were recorded on a Bruker Avance (400 MHz) (Bruker, Billerica, MA, USA), country and Jeol Avance (500 MHz) (Jeol Ltd., Tokyo, Japan). Chemical shifts are reported as δ values in parts per million (ppm) relative to tetramethylsilane or the residual solvent signals in DMSO-d6 or CDCl_3_ as the internal standard for the NMR spectra. Data are reported as follows: Chemical shift (multiplicity, coupling constants in Hz, integrated intensity, assignment). Abbreviations of multiplicity were as follows: s: Singlet, d: Doublet, t: Triplet, q: Quartet, m: Multiplet, br: Broad. High-resolution mass spectra (HRMS) data were obtained with a Micro-TOF mass spectrometer (Bruker, Billerica, MA, USA). IR spectra were recorded using the Thermo Scientific™ Nicolet™ iS50 FTIR spectrometer (Thermo Fisher Scientific, Waltham, MA, USA) with an ATR module and are reported in wavenumber (cm^−1^). Reactions were monitored by thin-layer chromatography (TLC) using aluminum Merck TLC plates coated with silica gel 60 F254. Normal-phase column chromatography was performed using silica gel 60 (0.063–0.200 mm, 70–230 mesh ASTM, Merck, Darmstadt, Germany). Melting points were measured using a melting point apparatus (Griffin) and are uncorrected.

### 3.2. Synthesis of Compounds ***1a***–***1g***

Compounds **1a**–**1g** were synthesized using a modified procedure [22]. Substituted isatoic anhydride (1.0 equiv.) was added to a round-bottom flask. DIPEA (2.0 equiv.) and dimethylacetamide (DMAC) were added at room temperature and stirred for 10 min. The flask was fitted with a rubber septum and purged with nitrogen gas. Appropriate alkyl halide (2.0 equiv.) was added dropwise, and the reaction was heated at 45 °C for 2 h and monitored by thin-layer chromatography. Upon completion, the reaction mixture was quenched with H_2_O. The precipitate was filtered, washed with H_2_O, and concentrated in vacuo to give the product.

*1-Methyl-2H-benzo[d] [1,3] oxazine-2,4(1H)-dione (***1a***)*: Isatoic anhydride (2.00 g, 12.3 mmol), methyl iodide (1.60 mL, 24.6 mmol), and DMAC (13 mL) were used to give the title compound (1.75 g, 81% yield) as an off-white solid. ^1^H NMR (400 MHz, DMSO) δ 8.01 (d, *J* = 7.7 Hz, 1H), 7.86 (t, *J* = 7.9 Hz, 1H), 7.44 (d, *J* = 8.4 Hz, 1H), 7.34 (t, *J* = 7.6 Hz, 1H), 3.47 (s, 3H); ^13^C NMR (101 MHz, DMSO) δ 158.95, 147.71, 142.18, 137.15, 129.28, 123.54, 114.80, 111.49, 31.63 [29].

*6-Methoxy-1-methyl-2H-benzo[d] [1,3] oxazine-2,4(1H)-dione (***1b***)*: 5-Methoxyisatoic anhydride (290 mg, 1.5 mmol), methyl iodide (186 μL, 3.0 mmol), and DMAC (3 mL) were used to give the title compound (191 mg, 61% yield) as a light-green solid. ^1^H NMR (400 MHz, DMSO) δ 7.52–7.34 (m, *J* = 17.7, 8.9 Hz, 3H), 3.83 (s, 3H), 3.45 (s, 3H); ^13^C NMR (101 MHz, DMSO) δ 158.84, 155.17, 147.52, 136.31, 124.99, 116.54, 112.12, 110.88, 55.80, 31.71 [30].

*7-Chloro-1-methyl-2H-benzo[d] [1,3] oxazine-2,4(1H)-dione (***1c***)*: 4-Chloroisatoic anhydride (300 mg, 1.5 mmol), methyl iodide (186 μL, 3.0 mmol), and DMAC (2 mL) were used to give the title compound (260 mg, 81% yield) as a white solid. ^1^H NMR (400 MHz, DMSO) δ 8.00 (d, *J* = 8.4 Hz, 1H), 7.58 (s, 1H), 7.39 (d, *J* = 8.3 Hz, 1H), 3.46 (s, 3H); ^13^C NMR (101 MHz, DMSO) δ 158.27, 143.39, 141.89, 131.01, 130.73, 123.64, 114.86, 110.59, 31.91 [31].

*1-Methyl-7-nitro-2H-benzo[d][1,3] oxazine-2,4(1H)-dione (***1d***)*: 4-Nitroisatoic anhydride (312 mg, 1.5 mmol), methyl iodide (186 μL, 3.0 mmol), and DMAC (2 mL) were used to give the title compound (306 mg, 81% yield) as a yellow solid. ^1^H NMR (400 MHz, DMSO) δ 8.25 (d, *J* = 8.5 Hz, 1H), 8.12 (d, *J* = 1.5 Hz, 1H), 8.06 (dd, *J* = 8.5, 1.8 Hz, 1H), 3.56 (s, 3H); ^13^C NMR (101 MHz, DMSO) δ 157.87, 152.43, 147.33, 143.06, 131.13, 117.50, 110.82, 109.84, 32.08 [30].

*6-Chloro-1-methyl-2H-benzo[d] [1,3] oxazine-2,4(1H)-dione (***1e***)*: 5-Chloroisatoic anhydride (297 mg, 1.5 mmol), methyl iodide (186 μL, 3.0 mmol), and DMAC (2 mL) were used to give the title compound (258 mg, 81% yield) as a white solid. ^1^H NMR (400 MHz, CDCl_3_) δ 8.22 (d, *J* = 2.4 Hz, 1H), 7.82 (dd, *J* = 8.9, 2.5 Hz, 1H), 7.36 (s, 1H), 3.68 (s, 3H); ^13^C NMR (101 MHz, CDCl_3_) δ 157.43, 147.61, 140.72, 137.37, 131.56, 130.16, 115.59, 113.11, 32.20 [32].

*6-Bromo-1-methyl-2H-benzo[d] [1,3] oxazine-2,4(1H)-dione (***1f***)*: 5-Bromoisatoic anhydride (363 mg, 1.5 mmol), methyl iodide (186 μL, 3.0 mmol), and DMAC (2 mL) were used to give the title compound (368 mg, 96% yield) as a brown solid. ^1^H NMR (400 MHz, CDCl_3_) δ 8.26 (d, *J* = 2.3 Hz, 1H), 7.85 (dd, *J* = 8.8, 2.3 Hz, 1H), 7.08 (d, *J* = 8.9 Hz, 1H), 3.57 (s, 3H); ^13^C NMR (101 MHz, CDCl_3_) δ 157.31, 147.59, 141.17, 140.19, 133.20, 117.06, 115.80, 113.42, 32.17 [29].

*1-Ethyl-2H-benzo[d] [1,3] oxazine-2,4(1H)-dione (***1g***)*: Isatoic anhydride (245 mg, 1.3 mmol), ethyl iodide (240 μL, 3.0 mmol), and DMAC (2 mL) were used to give the title compound (246 mg, 48% yield) as an off-white solid. ^1^H NMR (400 MHz, DMSO) δ 8.02 (d, *J* = 7.8 Hz, 1H), 7.85 (t, *J* = 7.9 Hz, 1H), 7.50 (d, *J* = 8.5 Hz, 1H), 7.33 (t, *J* = 7.6 Hz, 1H), 4.06 (q, *J* = 7.0 Hz, 2H), 1.23 (t, *J* = 7.1 Hz, 3H); ^13^C NMR (101 MHz, DMSO) δ 145.61, 141.10, 137.23, 129.61, 123.47, 122.62, 114.58, 111.75, 31.67, 11.86 [33].

### 3.3. Synthesis of Compounds ***2a*** and ***2b***

Compounds **2a** and **2b** were synthesized using a modified procedure [23]. Ethyl potassium malonate (2.0 equiv.), MgCl_2_ (3.0 equiv.), and NEt_3_ (3.0 equiv.) were added to a round-bottom flask. The flask was fitted with a rubber septum and purged with nitrogen gas, and dry MeCN was then added. The mixture was stirred at room temperature for 2 h, appropriate acid chloride (1.0 equiv.) was then added dropwise at 0 °C, and the mixture was stirred overnight at room temperature. Upon completion, the reaction mixture was extracted with CH_2_Cl_2_ and H_2_O. The combined organic layers were washed with brine, dried over anh. Na_2_SO_4_, filtered, and then concentrated in vacuo. The crude mixture was purified by silica gel column chromatography to give the product.

*Ethyl 3-oxohexadecanoate (***2a***)*: Ethyl potassium malonate (6.30 g, 37.0 mmol), myristoyl chloride (4.80 mL, 17.7 mmol), and MeCN (84 mL) were used to give the title compound (4.24 g, 80% yield) as a light-yellow oil. ^1^H NMR (400 MHz, CDCl_3_) δ 4.19 (q, *J* = 7.1 Hz, 2H), 3.42 (s, 2H), 2.52 (t, *J* = 7.4 Hz, 2H), 1.65–1.52 (m, 3H), 1.24 (s, 20H), 0.89–0.85 (m, *J* = 6.6 Hz, 5H); ^13^C NMR (101 MHz, CDCl_3_) δ 203.11, 167.43, 89.09, 61.46, 49.45, 43.19, 35.20, 33.97, 32.05, 29.80, 29.77, 29.72, 29.57, 29.48, 29.37, 24.86, 22.81, 14.22 [34].

*Ethyl 3-oxododecanoate (***2b***)*: Ethyl potassium malonate (1.7 g, 10 mmol), decanoyl chloride (1.6 mL, 7.5 mmol), and MeCN (23 mL) were used to give the title compound (1.271 g, 75% yield) as a light-yellow oil. ^1^H NMR (400 MHz, CDCl_3_) δ 4.18 (dt, *J* = 6.8, 5.4 Hz, 2H), 3.42 (s, 2H), 2.52 (t, *J* = 7.4 Hz, 2H), 1.60 (dd, *J* = 21.9, 14.9 Hz, 3H), 1.44–1.12 (m, 14H), 0.87 (t, *J* = 6.7 Hz, 3H); ^13^C NMR (101 MHz, CDCl_3_) δ 61.45, 49.45, 43.18, 35.20, 31.99, 29.53, 29.48, 29.37, 29.19, 29.16, 26.39, 23.62, 22.78, 14.23 [35].

### 3.4. Synthesis of Compounds ***3a***–***3j***, ***4d***, and ***5d***

Compounds **3a**–**3j**, **4d**, and **5d** were synthesized using a modified procedure [24]. Sodium hydride (60% in mineral oil) was added to a round-bottom flask and washed with hexane 3 times. The flask was fitted with a rubber septum and purged with nitrogen gas, and *N*-alkyl substituted isatoic anhydride (**1a**–**1g**), β-ketoester (**2a**–**2b**), and dry DMF were then added. The mixture was stirred overnight at room temperature. Upon completion, the reaction mixture was quenched with H_2_O. The resulting mixture was extracted with ethyl acetate (EtOAc). The combined organic layers were washed with brine, dried over anh. Na_2_SO_4_, filtered, and then concentrated in vacuo. The crude mixture was purified by silica gel column chromatography to give the products.

*4-Hydroxy-1-methyl-3-tetradecanoylquinolin-2(1H)-one (***3a***)*: **1a** (354 mg, 1.0 mmol), **2a** (596 mg, 1.0 mmol), NaH (60 mg, 1.5 mmol), and DMF (4 mL) were used and purified by silica gel column chromatography (eluent: 5% EtOAc in hexanes) to give the title product (186 mg, 24% yield) as an off-white solid. ^1^H NMR (400 MHz, CDCl_3_) δ 8.23 (d, *J* = 6.9 Hz, 1H), 7.72–7.66 (m, 1H), 7.30 (d, *J* = 8.6 Hz, 1H), 7.32–7.27 (m, 1H), 3.65 (s, 3H), 3.29 (t, *J* = 7.4 Hz, 2H), 1.76–1.67 (m, 2H), 1.57 (s, 2H), 1.28 (d, *J* = 16.6 Hz, 18H), 0.88 (t, *J* = 6.8 Hz, 3H); ^13^C NMR (101 MHz, CDCl_3_) δ 209.71, 174.28, 161.65, 152.68, 141.81, 134.90, 126.34, 124.24, 122.15, 115.84, 114.29, 57.11, 43.11, 32.08, 29.84, 29.80, 29.72, 29.71, 29.54, 29.51, 29.29, 24.49, 22.84, 14.25; IR (neat): 2916.00, 2848.35, 1651.82, 1624.97, 1590.46, 1557.51, 1502.15, 1463.21, 1427.31, 1409.30, 1339.12, 1303.63, 1280.61, 1251.95, 1198.68, 1171.19, 1061.74, 1039.80, 1030.70, 980.98; HRMS (ESI^+^): *m*/*z* calcd for C_24_H_34_NO_3_Na_2_^+^ [M− H + 2Na]^+^ 430.2329, found 430.2319; Mp: 70.0–70.8 °C.

*4-Hydroxy-6-methoxy-1-methyl-3-tetradecanoylquinolin-2(1H)-one (***3b***)*: **1b** (62 mg, 0.30 mmol), **2a** (90 mg, 0.30 mmol), NaH (18 mg, 0.45 mmol), and DMF (1 mL) were used and purified by silica gel column chromatography (eluent: EtOAc/hexanes = 1:4) to give the title product (21 mg, 22% yield) as a light-green solid. ^1^H NMR (400 MHz, CDCl_3_) δ 7.62 (d, *J* = 2.8 Hz, 1H), 7.31 (dd, *J* = 9.2, 2.9 Hz, 1H), 7.25 (d, *J* = 13.9 Hz, 1H), 3.90 (s, 3H), 3.64 (s, 3H), 3.31 (t, *J* = 7.4 Hz, 2H), 1.75–1.70 (m, 2H), 1.27 (s, 20H), 0.89 (t, *J* = 6.7 Hz, 3H); ^13^C NMR(101 MHz, CDCl_3_) δ 209.84, 173.65, 161.27, 154.90, 152.80, 136.63, 124.67, 122.63, 115.87, 106.59, 102.05, 89.45, 55.95, 43.11, 32.08, 29.84, 29.80, 29.72, 29.71, 29.54, 29.50, 29.40, 24.49, 22.84, 14.25. IR (neat): 2916.19, 2849.20, 1647.85, 1592.93, 1572.55, 1505.80, 1466.05, 1452.89, 1413.22, 1382.45, 1357.95, 1339.80, 1236.89, 1196.42, 1177.76, 1148.43, 1059.87, 1034.63, 1004.25, 857.67, 821.28; HRMS (ESI^+^): *m*/*z* calcd for C_25_H_36_NO_4_Na_2_^+^ [M − H + 2Na]^+^ 460.2434, found 460.2427; Mp: 67.8–71.5 °C.

*7-Chloro-4-hydroxy-1-methyl-3-tetradecanoylquinolin-2(1H)-one (***3c***)*: **1c** (127 mg, 0.60 mmol), **2a** (150 mg, 0.50 mmol), NaH (30 mg, 0.75 mmol), and DMF (1 mL) were used and purified by silica gel column chromatography (eluent: 4% EtOAc in hexanes) to give the title product (26 mg, 12% yield) as a white solid. ^1^H NMR (400 MHz, CDCl_3_) δ 8.15 (d, *J* = 8.6 Hz, 1H), 7.30 (d, *J* = 1.2 Hz, 1H), 7.22 (dd, *J* = 8.6, 1.5 Hz, 1H), 3.61 (s, 3H), 3.27 (t, *J* = 7.4 Hz, 2H), 1.75–1.65 (m, 2H), 1.26 (s, 20H), 0.88 (t, *J* = 6.8 Hz, 3H); ^13^C NMR (101 MHz, CDCl_3_) δ 209.63, 173.81, 171.52, 161.49, 142.53, 141.38, 127.71, 122.72, 114.36, 105.86, 105.60, 97.19, 86.07, 43.04, 32.08, 29.83, 29.79, 29.71, 29.68, 29.51, 29.41, 24.43, 22.83, 14.24; IR (neat): 2916.03 2852.38, 1644.46, 1609.69, 1556.11, 1473.10, 1443.16, 1421.20, 1378.16, 1343.22, 1320.45, 1264.80, 1227.80, 1203.14, 1099.25, 1031.70, 966.59, 891.66, 841.66; HRMS (ESI^+^): *m*/*z* calcd for C_24_H_33_ClNO_3_^−^ [M − H]^−^ 418.2154, found 418.2148; Mp: 103.4–103.9 °C.

*4-Hydroxy-1-methyl-7-nitro-3-tetradecanoylquinolin-2(1H)-one (***3d***)*, ethyl 1-methyl-7-nitro-4-oxo-2-tridecyl-1,4-dihydroquinoline-3-carboxylate (**4d**), and 2-(methylamino)-4-nitrobenzoic acid (**5d**): **1d** (134 mg, 0.60 mmol), **2a** (150 mg, 0.50 mmol), NaH (30 mg, 0.75 mmol), and DMF (1 mL) were used and purified by silica gel column chromatography (eluent: 4% EtOAc in hexanes) to give **3d** (18 mg, 9% yield) as a yellow solid, **4d** (40 mg, 17% yield) as a yellow-green solid, and **5d** (50 mg, 51% yield) as an orange solid, respectively.

**3d**: ^1^H NMR (400 MHz, CDCl_3_) δ 8.40 (d, *J* = 8.8 Hz, 1H), 8.17 (d, *J* = 1.8 Hz, 1H), 8.05 (dd, *J* = 8.7, 1.8 Hz, 1H), 3.72 (s, 3H), 3.30 (t, *J* = 7.4 Hz, 2H), 1.71 (d, *J* = 7.3 Hz, 2H), 1.43–1.37 (m, 2H), 1.26 (s, 18H), 0.88 (t, *J* = 6.7 Hz, 3H); ^13^C NMR (101 MHz, CDCl_3_) δ 210.00, 172.68, 161.20, 141.89, 132.97, 128.08, 120.13, 116.14, 111.27, 110.23, 109.66, 52.31, 46.67, 43.24, 32.07, 29.82, 29.79, 29.68, 29.65, 29.50, 29.45, 24.31, 22.83, 14.24. IR (neat): 2920.29, 2852.03, 1630.31, 1605.71, 1563.83, 1519.60, 1470.98, 1346.51, 1321.48, 1278.93, 1277.83, 1104.60, 1033.57, 986.91, 888.50, 828.10, 813.47; HRMS (ESI^+^): *m*/*z* calcd for C_24_H_33_N_2_O_5_^−^ [M − H]^−^ 429.2395, found 429.2384; Mp: 156.3–168.4 °C.

**4d**: ^1^H NMR (400 MHz, CDCl_3_) δ 8.55 (d, *J* = 8.7 Hz, 1H), 8.45 (s, 1H), 8.13 (d, *J* = 8.7 Hz, 1H), 4.41 (d, *J* = 7.1 Hz, 2H), 3.86 (s, 3H), 2.87–2.71 (m, 2H), 1.71 (d, *J* = 7.0 Hz, 2H), 1.44 (d, *J* = 7.0 Hz, 2H), 1.39 (t, *J* = 7.1 Hz, 3H), 1.25 (s, 18H), 0.87 (t, *J* = 6.7 Hz, 3H); (101 MHz, CDCl_3_) δ 173.30, 166.95, 154.84, 150.20, 141.36, 129.85, 129.18, 120.11, 117.84, 112.11, 61.87, 35.23, 32.60, 32.02, 29.79, 29.76, 29.73, 29.67, 29.56, 29.44, 29.28, 29.04, 22.79, 14.37. IR (neat): 2917.90, 2852.52, 1726.40, 1597.66, 1518.79, 1496.50, 1464.33, 1384.73, 1345.48, 1214.81, 1135.89, 1105.59, 1037.28, 1022.98, 890.49, 855.73, 825.02; HRMS (ESI^+^): *m*/*z* calcd for C_26_H_38_N_2_O_5_Na^+^ [M + Na]^+^ 481.2678, found 481.2676; Mp: 92.6–95.2 °C.

**5d**: ^1^H NMR (400 MHz, DMSO) δ 7.99 (d, *J* = 8.6 Hz, 1H), 7.37 (d, *J* = 1.7 Hz, 1H), 7.33–7.27 (m, 1H), 3.46 (s, 1H), 2.91 (s, 3H); ^13^C NMR (101 MHz, DMSO) δ 168.63, 151.78, 151.39, 133.26, 115.08, 107.65, 104.81, 29.32 [36].

*6-Chloro-4-hydroxy-1-methyl-3-tetradecanoylquinolin-2(1H)-one (***3e***)*: **1e** (159 mg, 0.75 mmol), **2a** (150 mg, 0.50 mmol), NaH (30 mg, 0.75 mmol), and DMF (1 mL) were used and purified by silica gel column chromatography (eluent: 5% EtOAc in hexanes) to give the title product (60 mg, 28% yield) as a white solid. ^1^H NMR (400 MHz, CDCl_3_) δ 8.21 (d, *J* = 2.4 Hz, 1H), 7.65 (dd, *J* = 9.0, 2.5 Hz, 1H), 7.29 (d, *J* = 3.2 Hz, 1H), 3.66 (s, 3H), 3.31 (t, *J* = 7.4 Hz, 2H), 1.73 (dt, *J* = 14.9, 7.5 Hz, 2H), 1.64 (s, 2H), 1.42 (d, *J* = 8.8 Hz, 2H), 1.28 (s, 16H), 0.91 (t, *J* = 6.8 Hz, 3H); ^13^C NMR (101 MHz, CDCl_3_) δ 209.76, 173.14, 161.26, 140.24, 134.85, 130.85, 128.03, 125.56, 116.91, 115.86, 65.84, 57.63, 47.37, 43.11, 32.07, 29.83, 29.79, 29.70, 29.68, 29.50, 29.46, 24.39, 22.83, 14.24. IR (neat): 2919.88, 2847.72, 1651.26, 1635.04, 1610.19, 1551.49, 1463.26, 1402.46, 1293.20, 1206.95, 1093.57, 1052.40, 955.38, 897.40, 819.76, 812.24; HRMS (ESI^+^): *m*/*z* calcd for C_24_H_33_ClNO_3_^−^ [M − H]^−^ 418.2154, found 418.2149; Mp: 73.5–80.0 °C.

*6-Bromo-4-hydroxy-1-methyl-3-tetradecanoylquinolin-2(1H)-one (***3f***)*: **1f** (192 mg, 0.75 mmol), **2a** (150 mg, 0.50 mmol), NaH (30 mg, 0.75 mmol), and DMF (1 mL) were used and purified by silica gel column chromatography (eluent: 5% EtOAc in hexanes) to give the title product (63 mg, 27% yield) as a light-brown solid. ^1^H NMR (400 MHz, CDCl_3_) δ 8.31 (d, *J* = 2.1 Hz, 1H), 7.74 (dd, *J* = 9.0, 2.3 Hz, 1H), 7.17 (d, *J* = 9.0 Hz, 1H), 3.61 (s, 3H), 3.27 (t, *J* = 7.4 Hz, 2H), 1.70 (dt, *J* = 14.9, 7.4 Hz, 2H), 1.40 (s, 2H), 1.25 (s, 18H), 0.88 (d, *J* = 6.5 Hz, 3H); ^13^C NMR (101 MHz, CDCl_3_) δ 209.71, 173.06, 161.21, 140.61, 137.54, 128.63, 117.29, 116.07, 115.17, 106.28, 61.69, 56.13, 55.98, 43.09, 32.06, 29.82, 29.78, 29.69, 29.67, 29.49, 29.41, 24.37, 22.82, 14.23. IR (neat): 2919.92, 2851.49, 1644.22, 1606.11, 1551.76, 1494.12, 1464.02, 1429.07, 1413.33, 1317.00, 1292.07, 1252.78, 1149.12, 1062.14, 1000.11, 821.17, 807.49; HRMS (ESI^+^): *m*/*z* calcd for C_24_H_33_BrNO_3_^−^ [M − H]^−^ 462.1649, found 462.1657; Mp: 73.2–75.0 °C.

*1-Ethyl-4-hydroxy-3-tetradecanoylquinolin-2(1H)-one (***3g***)*: **1g** (144 mg, 0.75 mmol), **2a** (150 mg, 0.50 mmol), NaH (30 mg, 0.75 mmol), and DMF (1 mL) were used and purified by silica gel column chromatography (eluent: 5% EtOAc in hexanes) to give the title product (16 mg, 8% yield) as an off-white solid. ^1^H NMR (400 MHz, CDCl_3_) δ 8.24 (dd, *J* = 8.0, 1.3 Hz, 1H), 7.71–7.62 (m, 1H), 7.31 (d, *J* = 8.6 Hz, 1H), 7.23 (d, *J* = 7.3 Hz, 1H), 4.60–4.00 (m, 2H), 3.70–3.02 (m, 3H), 1.70 (dt, *J* = 23.8, 11.5 Hz, 2H), 1.41–1.10 (m, 23H), 0.87 (t, *J* = 6.8 Hz, 3H); ^13^C NMR (101 MHz, CDCl_3_) δ 209.71, 174.07, 161.16, 140.88, 134.83, 133.69, 126.53, 123.77, 121.92, 116.00, 114.14, 105.95, 43.16, 37.22, 32.07, 29.83, 29.79, 29.71, 29.51, 29.50, 29.18, 24.41, 22.83, 14.24, 12.94; *m*/*z* calcd for C_25_H_37_NO_3_^−^ [M + 3H]^+^ 402.3008, found 402.3003; Mp: 74.1–75.5 °C.

*3-Decanoyl-4-hydroxy-1-methylquinolin-2(1H)-one (***3h***)*: **1a** (266 mg, 1.50 mmol), **2b** (365 mg, 1.50 mmol), NaH (90 mg, 2.25 mmol), and DMF (1 mL) were used and the crude product was purified by silica gel column chromatography (eluent: 5% EtOAc in Hexanes) to give the title product (35 mg, 7% yield) as a light-yellow oil. ^1^H NMR (500 MHz, CDCl_3_) δ 8.25–8.08 (m, 1H), 7.65 (d, *J* = 7.1 Hz, 1H), 7.24 (dd, *J* = 16.2, 8.1 Hz, 2H), 3.61 (dd, *J* = 15.7, 1.4 Hz, 3H), 3.25 (d, *J* = 7.1 Hz, 2H), 1.68 (d, *J* = 6.2 Hz, 2H), 1.25 (s, 13H), 0.84 (d, *J* = 6.2 Hz, 3H); ^13^C NMR (126 MHz, CDCl_3_) δ 209.70, 174.24, 161.64, 141.75, 134.94, 126.31, 122.19, 115.78, 114.32, 105.95, 43.17, 32.07, 29.73, 29.70, 29.54, 29.48, 29.30, 24.46, 22.85, 14.29; *m*/*z* calcd for C_20_H_27_NO_3_^−^ [M − H]^−^ 374.1708, found 374.1711; Mp: 65.0–69.1 °C.

*6-Chloro-3-decanoyl-4-hydroxy-1-methylquinolin-2(1H)-one (***3i***)*: **1e** (80 mg, 0.38 mmol), **2b** (92 mg, 0.38 mmol), NaH (22.8 mg, 0.57 mmol), and DMF (1 mL) were used and purified by silica gel column chromatography (eluent: 5% EtOAc in Hexanes) to give the title product (31 mg, 22% yield) as an off-white solid. ^1^H NMR (500 MHz, CDCl_3_) δ 8.57–7.97 (m, 1H), 7.82–7.43 (m, 1H), 7.53–6.99 (m, 1H), 3.63 (d, *J* = 13.9 Hz, 3H), 3.34–3.16 (m, 2H), 1.71 (s, 2H), 1.29 (s, 13H), 0.88 (d, *J* = 6.1 Hz, 3H). ^13^C NMR (126 MHz, CDCl_3_) δ 209.80, 173.15, 161.29, 140.23, 134.90, 128.05, 125.58, 116.89, 115.92, 107.49, 106.33, 43.19, 32.08, 29.72, 29.70, 29.51, 29.49, 24.39, 22.86, 14.30; *m*/*z* calcd for C_20_H_25_ClNO_3_^−^ [M − H]^−^ 362.1523, found 362.9271; Mp: 79.5–81.0 °C.

*6-Bromo-3-decanoyl-4-hydroxy-1-methylquinolin-2(1H)-one (***3j***)*: **1f** (139 mg, 0.60 mmol), **2b** (150 mg, 0.60 mmol), NaH (36 mg, 0.90 mmol), and DMF (1 mL) were used and purified by silica gel column chromatography (eluent: 5% EtOAc in Hexanes) to give the title product (18 mg, 7% yield) as a light-brown solid. ^1^H NMR (500 MHz, CDCl_3_) δ 8.40 (d, *J* = 15.5 Hz, 1H), 7.80 (d, *J* = 8.6 Hz, 1H), 7.24 (d, *J* = 8.9 Hz, 1H), 3.69 (d, *J* = 15.0 Hz, 3H), 3.34 (d, *J* = 7.4 Hz, 2H), 1.78 (d, *J* = 5.8 Hz, 2H), 1.35 (s, 13H), 0.94 (d, *J* = 6.8 Hz, 3H); ^13^C NMR (126 MHz, CDCl_3_) δ 209.76, 173.06, 161.25, 140.61, 137.61, 128.99, 128.64, 117.28, 116.15, 115.22, 106.30, 43.18, 32.08, 29.70, 29.49, 24.83, 24.38, 22.86, 20.96, 14.28; *m*/*z* calcd for C_20_H_27_BrNO_3_^−^ [M + H]^+^ 408.1174, found 408.1335; Mp: 75.0–76.8 °C.

### 3.5. Evaluation of Antibacterial Activity

The antibacterial activity method was modified from the Manual of Antimicrobial Susceptibility Testing, American Society of Microbiology [37]. The minimum inhibitory concentrations (MICs) of each synthesized quinolone against *Staphylococcus aureus* (ATCC 6538P) and *Escherichia coli* (ATCC 25922) were determined using resazurin-96 well plate microdilution assay. Briefly, the quinolone solutions were prepared by dissolving the synthesized compounds in 4% *v*/*v* dimethyl sulfoxide (DMSO) in Mueller-Hinton Broth (MHB) to obtain the concentrations of 2000, 1000, 500, 250, 125, and 62.5 µg/mL. Ciprofloxacin, an antibiotic in the fluoroquinolone class, was used as a positive control. The 4% MHB solutions with and without bacteria were also used as experimental controls. The bacterial cultures were prepared by growing each strain in MHB at 37 °C for 3–5 h. Then, bacterial densities were measured by spectrometry at OD 600 nm (Genesys 20, Thermo Scientific, Waltham, MA, USA) and adjusted the cell concentration to 1.15 × 106 cells/mL. To test the antibacterial activity, 50 mL of each prepared quinolone solution was added into each well of 96-well plates, following by addition of 50 mL bacterial solution, so the final concentrations of the tested compounds were 1000, 500, 250, 125, 62.5, and 31.25 µg/mL. Then, the 96-well plates were closed, wrapped with parafilm to prevent the dehydration, and incubated at 37 °C for 21 h. After incubating for 21 h, 30 mL of 0.015 *w*/*v* resazurin aqueous solution was added into each well followed by 37 °C incubation for 1 h. The lowest concentrations of the agents that displayed blue color (no color change of resazurin) were assigned as MIC values.

In order to investigate whether the synthesized quinolones could kill the bacteria or not, the minimum bactericidal concentration (MBC) tests were performed by streaking the contents obtained from the wells appearing in a blue color on a Nutrient Agar (NA) medium and incubating at 37 °C for 24 h. The MBC values were defined as the lowest concentrations that the bacteria could not grow on the culture medium.

### 3.6. Evaluation of Antifungal Activity

The broth microdilution method for non-dermatophyte molds was performed to observe the antifungal activity using 96-well plates in RPMI media according to CLSI M38 (2017) [38]. In addition, 1 × 10^4^ spores of *Aspergillus flavus* ATCC204304 were utilized and co-incubated with amphotericin B, a fungicidal antifungal agent, in serial dilutions, i.e., 0.125–8 µg/mL (final concentrations), and different quinolone agents dissolved in DMSO in serial dilutions, i.e., 0.39–100 µg/mL (final concentrations), at 37 °C. XTT (sodium 2,3-bis(2-methoxy-4-nitro-5-sulfophenyl)-5-[(phenylamino)-carbonyl]-2H-tetrazolium) solution (0.5 mg/mL in PBS) was then added into each well to observe the growth of *A. flavus* ATCC204304 at 48 h [39]. The plate was further incubated for 15 min at 37 °C and the supernatant was collected to measure the OD at 490 nm using a spectrophotometer (Lambda 1050+ UV/Vis/NIR, PerkinElmer, Waltham, MA, USA). The half maximal inhibitory concentration (IC_50_) of each quinolone agent was then calculated using linear regression analyses. The experiments were performed in biological triplicates.

## 4. Conclusions

In summary, ten 3-alkyl-4-hydroxy-2-quinolone analogs (**3a**–**3j**), as well as the isomer **4d**, were synthesized and fully characterized. Most of the compounds showed substantial antifungal activity against *A. flavus*. The introduction of substituents, especially for halogens, at C-6 and C-7 led to a significant increase in the antifungal activity. The activity was also enhanced when replacing the tridecyl group at C-3 with the nonyl group. The SAR knowledge obtained from this study might be crucial for the development of new antimicrobial agents based on the 4-hydroxy-2-quinolone privilege scaffold.

## Figures and Tables

**Figure 1 molecules-25-03059-f001:**
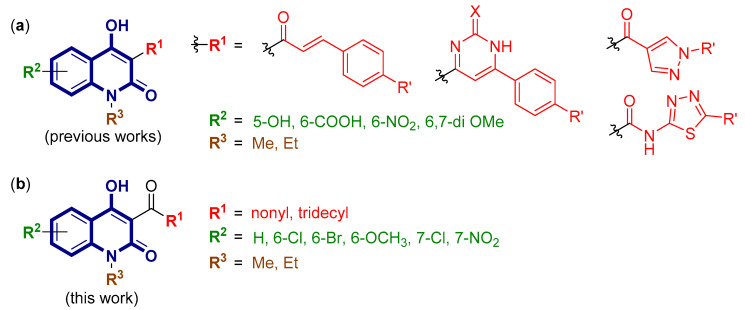
4-Hydroxy-2-quinolone analogs: (**a**) Examples of the structure–activity relationship investigated in previous works; (**b**) novel 4-hydroxy-2-quinolone analogs bearing alkyl side chain at C-3 and various types of substituents on C-6 and C-7 positions (this work).

**Figure 2 molecules-25-03059-f002:**
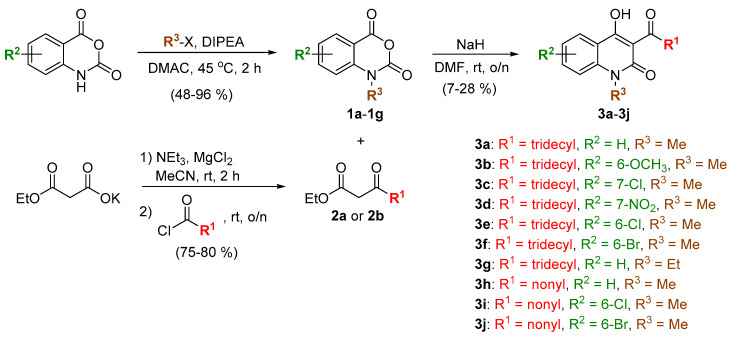
Synthetic scheme of the 3-step protocol for the preparation of 4-hydroxy-2-quinolone analogs (**3a**–**3j**).

**Figure 3 molecules-25-03059-f003:**
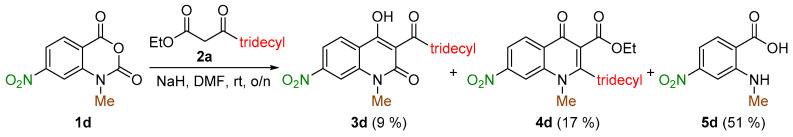
Synthesis of **3d** from the reaction between **1d** and **2a**, generating 2 additional byproducts, **4d** and **5d**.

**Table 1 molecules-25-03059-t001:** Antifungal and antibacterial activity of **3a**–**3j** and **4d**.

Compound	Substituents			IC_50_ ^1^ (µg/mL)	MIC ^2^ (µg/mL)	
	R ^1^	R ^2^	R ^3^	*A. flavus*	*S. aureus*	*E. coli*
**3a**	tridecyl	H	Me	70.97 ± 3.71	>1000	>1000
**3b**	tridecyl	6-OCH_3_	Me	42.15 ± 18.61	>1000	>1000
**3c**	tridecyl	7-Cl	Me	14.72 ± 2.35	>1000	>1000
**3d**	tridecyl	7-NO_2_	Me	38.45 ± 5.51	>1000	>1000
**3e**	tridecyl	6-Cl	Me	37.08 ± 1.52	>1000	>1000
**3f**	tridecyl	6-Br	Me	23.19 ± 3.19	>1000	>1000
**3g**	tridecyl	H	Et	9.48 ± 6.17	>1000	>1000
**3h**	nonyl	H	Me	8.45 ± 1.59	>1000	>1000
**3i**	nonyl	6-Cl	Me	4.60 ± 1.14	*	>1000
**3j**	nonyl	6-Br	Me	1.05 ± 1.31	**	>1000
**4d** ^3^	tridecyl ^3^	7-NO_2_ ^3^	Me ^3^	23.44 ± 1.63	>1000	>1000
AMB ^4^	–	–	–	1.93 ± 1.04	ND	ND
CIP ^5^	–	–	–	ND	0.835	0.0288

^1^ Half maximal inhibitory concentration (IC_50_) of the antifungal activity against *A. flavus* determined by broth microdilution method, expressed as mean ± SD; ^2^ Minimum inhibitory concentration (MIC) of the antibacterial activity against *S. aureus* and *E. coli* determined by broth microdilution method. All experiments were performed in biological triplicates; ^3^ The structure is based on the 1,4-dihydroquinoline-3-carboxylate scaffold (see Figure 3); ^4^ AMB = amphotericin B as a positive control for the antifungal activity; ^5^ CIP = ciprofloxacin as a positive control for the antibacterial activity; ND = No data. The MIC values of **3i** (*) and **3j** (**) were unclear; however, both compounds showed inhibitory activities at the concentrations between 125–1000 and 125–500 µg/mL, respectively (Appendix A).

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
