# Peer review of "Synthesis and Antimicrobial Activity of Novel 4-Hydroxy-2-quinolone Analogs"

_molecules, 2020, doi:10.3390/molecules25133059_

Round 1

Reviewer 1 Report

See atachment.

Reviewer 2 Report

I have reviewed the paper entitled "Synthesis and Antimicrobial Activity of Novel 2 4-Hydroxy-2-quinolone Analogues " First of all, comment that my knowledge in synthesis and structural determination of compounds is limited, then I have not delved into the methodology of this paper. If I would like to comment on some aspects:

 1.The title is not very appropriate. They do not have antimicrobial activity and they do have antifungal activity, that is not mentioned in the title.

  1. From my point of view in the introduction it would be necessary to add information. Why these compounds have been synthesized? Why these radicals have been selected? What particularity or biological activity is attributed to these radicals?
  2. I also find the results and discussion section deficient. It is more an exhibition of results than a discussion. It would be necessary to go deeper and add more information on the relationship between the activity of these compounds and their chemical structure.

Other aspects:

-In figure 1 the R3 is missing

-Materials and methods are missing "Evaluation of antifungal activity"

-Line 348: “The minimum inhibitory concentrations (MIC) of 348 each synthesized quinolone against Staphylococcus aureus (ATCC 6538P) and Escherichia coli” in italics

-In conclusions: "The most active compound 3j exhibited greater activity than the current antifungal medication, amphotericin B, and also showed moderate antibacterial activity against S. aureus" These results are not conclusive, they are doubtful, it should not be taken as conclusions.

Round 2

Reviewer 2 Report

I believe the manuscript has been improved and now warrants publication in Molecules